# Biomedical Interventions for HIV Prevention and Control: Beyond Vaccination

**DOI:** 10.3390/v17060756

**Published:** 2025-05-26

**Authors:** Yu Liao, Ziyu Wen, Minjuan Shi, Huachun Zou, Caijun Sun

**Affiliations:** 1School of Public Health (Shenzhen), Sun Yat-sen University, Shenzhen 518107, China; liaoy237@mail2.sysu.edu.cn (Y.L.); wenzy8@mail.sysu.edu.cn (Z.W.); shimj8@mail2.sysu.edu.cn (M.S.); 2Shenzhen Key Laboratory of Pathogenic Microbes and Biosafety, Shenzhen Campus, Sun Yat-sen University, Shenzhen 518107, China; 3Key Laboratory of Public Health Safety of the Ministry of Education, School of Public Health, Fudan University, Shanghai 200032, China; 4Key Laboratory of Tropical Disease Control, Sun Yat-sen University, Ministry of Education, Guangzhou 514400, China; 5State Key Laboratory of Anti-Infective Drug Discovery and Development, School of Pharmaceutical Sciences, Sun Yat-sen University, Guangzhou 510006, China

**Keywords:** HIV, AIDS epidemic, prevention, biomedical interventions, 2030 goal

## Abstract

The global HIV epidemic remains persistent, mostly because of neither a drug for its cure nor a vaccine for its prevention. An HIV vaccine is thought of as the most cost-effective biomedical intervention to eventually terminate the HIV epidemic, but it is not clinically available yet due to technical hurdles. However, beyond vaccination, increasing alternative and innovative biomedical interventions have been developed for the prevention and control of HIV infections. Herein, we discuss the challenges encountered in the innovation of biomedical interventions against HIV infections, and summarize the landscape and latest advances of these biomedical interventions to intercept the risk of HIV acquisition and transmission, aiming to provide valuable clues for exploring more out-of-the-box solutions to prevent HIV infections, thereby contributing to realizing the 2030 goal of ending the AIDS epidemic.

## 1. Introduction

More than 40 years have passed since the discovery of the human immunodeficiency virus (HIV) and acquired immunodeficiency syndrome (AIDS), yet neither a drug for its cure nor a vaccine for its prevention are available [1,2]. Despite advances in behavioral interventions, such as needle exchange programs, condom promotion, and partner education, the HIV epidemic remains a persistent global health challenge. As reported by the Joint United Nations Programme on HIV/AIDS (UNAIDS), there are currently 39.9 million people living with HIV (PLWH), and more than 40 million have died from AIDS-related complications [3]. In 2023, 1.3 million new infections were reported, underscoring the urgent need for more effective strategies [3]. With continuous advancements in HIV testing technologies and the increased availability of antiretroviral therapy (ART), 86% of people living with HIV (PLWH) were aware of their HIV status, and 30.7 million individuals (77% of PLWH) were receiving ART by the end of 2023.

However, the UNAIDS “90-90-90” global targets, proposed in 2014, have not yet been accomplished. These targets aimed to ensure the following by 2020: (1) 90% of PLWHs know their status, (2) 90% of diagnosed PLWHs receive sustained ART, and (3) 90% of PLWHs on ART treatment achieve viral suppression. Subsequently, the “95-95-95” targets for 2025—which demand 95% success rates for diagnosis, treatment initiation, and viral suppression—now also face significant challenges. The promotion of ART coverage—combined with biomedical interventions like pre-exposure prophylaxis (PrEP) and post-exposure prophylaxis (PEP)—can effectively reduce HIV transmission [4]. Due to the persistence of latent viral reservoirs in host cells, despite long-term treatment [5], and a rapid viral rebound after ART interruption [6], the current ART drugs alone cannot eliminate HIV infections, but ART treatment can render an epidemiologically irrelevant transmission risk under sustained virological suppression based on the Undetectable = Untransmittable (U=U) principle. Nevertheless, long-term use of ART drugs is associated with adverse events, such as cardiovascular disease and neurocognitive disorders [7,8,9]. Thus, ART therapy, rather offering an HIV cure, has transformed AIDS into a kind of chronic disease, which prompts new medical and social challenges [10,11]. In summary, the development of innovative biomedical interventions will be crucial for achieving the 2030 goal of ending the AIDS epidemic. While an effective HIV-preventive vaccine remains elusive due to technical hurdles, research efforts have increasingly shifted toward exploring alternative strategies. Herein, we provide an overview of the current landscape of HIV biomedical interventions beyond the HIV vaccine, highlighting their potential to mitigate the HIV epidemic (Figure 1). We aim to provide insights on exploring out-of-the-box solutions against HIV infections, and thus contribute to efforts for ending the AIDS epidemic.

## 2. Challenges for the Development of Biomedical Interventions Against HIV Acquisition and Transmission

To establish a foundation for evidence-based HIV biomedical interventions, it is necessary to have a comprehensive understanding of the difficulties and challenges in developing new strategies against HIV acquisition and transmission. Therefore, we have summarized these challenges encountered in the innovation of biomedical interventions against HIV infections (Figure 2).

### 2.1. High Genetic Variability

The extensive genetic variation in the HIV genome, driven by frequent mutations and recombination, represents a significant challenge in controlling the global HIV pandemic. As a retrovirus with a single-stranded RNA genome, HIV exhibits a mutation rate of 5 to 10 bases per replication cycle, attributable to the absence of proofreading by reverse transcriptase [12]. Additional errors introduced during transcription by RNA polymerase II [13] further contribute to the genetic diversity of HIV, particularly in the envelope (env) gene. This gene encodes proteins that are essential for viral entry into host cells. The envelope protein, a primary target of neutralizing antibodies, demonstrates substantial variability, which poses a significant obstacle to the development of broadly effective vaccines and antibodies [14]. Additionally, HIV undergoes frequent recombination, which significantly complicates efforts to prevent the transmission of this virus. HIV possesses two single-stranded, positive-sense RNA genomes. However, these two RNA strands are not identical copies and do not undergo conventional meiosis or homologous pairing during genetic recombination. Consequently, HIV cannot be classified as a true diploid, but is instead termed a “pseudodiploid” virus. The pseudodiploid nature of HIV arises may through the following two key mechanisms: (1) cellular co-infection by multiple viral particles, and (2) packaging of heterologous RNA genomes into individual virions [15,16]. This dual-RNA architecture enables a single infected cell to harbor multiple genomic templates. During reverse transcription, the viral polymerase undergoes frequent template switching between these RNA strands, generating recombinant proviral DNA with enhanced genetic diversity. This molecular recombination serves as a major driver of HIV’s evolutionary adaptability, facilitating immune evasion and drug resistance development [17]. During reverse transcription, reverse transcriptase can switch selectively between the two RNA template strands, producing a chimeric DNA molecule and resulting in genetic recombination, with an average frequency of up to five recombinant events per proviral genome synthesis [18].

The high recombination rates in HIV genes have contributed to the emergence of diverse subtypes, recombinant forms, and unique strains, exacerbating the global pandemic [19,20]. HIV can be divided into HIV-1 and HIV-2. HIV-1 and HIV-2 originate from different types of SIV in different monkey species. HIV-1 is closely related to SIV in chimpanzees (SIVcpz) [21], whereas HIV-2 is more closely related to SIV in white-browed monkeys (SIVsm) [22]. Compared to HIV-1, HIV-2 exhibits a significantly lower mutation rate, reduced pathogenicity, and diminished transmissibility. As a result, HIV-2 infections represent only a minor fraction of global HIV cases. Geographically, HIV-2 remains largely restricted to West Africa, likely due to a combination of epidemiological factors and limited spread beyond the region (if not specified, HIV in the following text refers to HIV-1). HIV-1 has evolved into four major groups, which are as follows: M, O, N, and P, with Group M further subdivided into eleven subtypes [23]. These four groups emerged from four independent zoonotic transmissions, with group M being the predominant and globally prevalent form of HIV-1 [24]. Amino acid differences within subtypes can range from 8 to 30%, and between subtypes, from 17 to 42%, greatly complicating the development of vaccines and other biomedical interventions [25].

Importantly, mutation and recombination in HIV are not independent, but interact synergistically. Some studies have shown that mutation rates are significantly higher in recombinant regions, with approximately 15–20% of mutations associated with genetic recombination [26]. These genetic variations complicate the design of interventions, necessitating broad-spectrum strategies to address the diversity of circulating strains. Under selective pressures from host immune responses and antiretroviral drugs, HIV evolves rapidly, making it highly adaptable to the host, and thus leading to significant challenges for prevention and control. The genetic diversity of HIV arises not only from the error-prone nature of reverse transcriptase and frequent recombination events, but also through the action of host-encoded mutagenic enzymes. The APOBEC3 family, particularly APOBEC3G and APOBEC3F, plays a prominent role in this process. When HIV fails to express functional Vif protein, these cytidine deaminases are packaged into virions and subsequently induce G-to-A hypermutations by deaminating cytosines in the minus-strand cDNA during reverse transcription [27]. While extensive APOBEC-mediated hypermutation generally proves to be lethal to the virus, sublethal levels of editing may allow viral variants to evade immune detection and become established in the population, thereby driving viral evolution and contributing to drug resistance [28]. Additional host factors, including ADARs (adenosine deaminases acting on RNA), may further modify HIV RNA through editing, although their exact role in shaping viral diversity requires further elucidation [29]. Consequently, biomedical interventions must account for the complexity of subtype variability, necessitating broad-spectrum strategies that encompass multiple subtypes.

### 2.2. Highly Glycosylated Antigens

Glycan arrays that densely surround the surface of the envelope antigen represent another distinctive feature of HIV. Glycosylation, a common post-translational modification of proteins, is closely linked to the proper folding and conformational changes of viral proteins [30]. Glycosylation can influence viral infectivity by facilitating viral attachment, cell fusion, and migration, and also plays a role in viral replication, including assembly and budding [31]. HIV gp120 glycoprotein has a highly glycosylated surface and is among the most heavily glycosylated non-synthetic proteins. A single envelope protein trimer contains more than 90 N-linked glycans, which comprise over half of its mass [32]. Compared to the binding of antibodies to protein antigens, the affinity of antibodies for carbohydrate-based antigens is relatively low [33], which impedes cross-linking and receptor-binding of B-cell receptors (BCRs). Regarding the source of glycans, HIV does not encode the gene product responsible for glycan synthesis. Instead, HIV utilizes host cell’s cellular enzymes and raw materials to glycosylate the surface of viral antigens. The antigen surface, covered by the host’s “self” glycans, enables HIV to develop immune tolerance to the functional antibodies and evade antibody recognition [34]. Furthermore, due to the homology between HIV glycans and host glycans, antibodies generated against the viral N-linked glycans may cross-react with the host’s N-linked glycans. Structurally, the highly glycosylated structure of the HIV antigen confers unique properties that significantly complicate HIV prevention. On one hand, this highly glycosylated structure can harbor key conserved antigenic epitopes of HIV, reducing the overall antigenicity of gp120 and making it exceptionally challenging for the immune system to generate the corresponding broadly neutralizing antibodies [35]. On the other hand, the conformational plasticity of glycosylated antigens enables them to undergo “conformational masking”, allowing them to escape recognition by neutralizing antibodies and thus protecting their vulnerable regions from immune attack [36]. For instance, although the receptor-binding region of gp120 lacks glycosylated modifications, the glycan located near the CD4 receptor-binding site creates a masking effect that renders it resistant to antibodies targeting this epitope, thereby evading antibody-mediated neutralization [37].

### 2.3. Latent Viral Reservoirs

After successfully invading host cells, HIV can synthesize viral DNA through the action of reverse transcriptase and integrate it into the host genome via the enzyme integrase. During this process, HIV can persist for extended periods in the form of pre-integrated viral DNA, particularly in long-lived cells, such as resting CD4+ memory T cells, forming what is known as a latent viral reservoir [38,39]. Conventional ART drugs are effective in targeting and eliminating free virions undergoing active replication. However, they have minimal impact on latent HIV reservoirs in a quiescent state. Even in patients receiving long-term ART with undetectable plasma viral loads, latent viral reservoirs persist [5]. While conventional ART effectively suppresses the plasma viral load to undetectable levels and prevents productive virions from productively infecting cells, viral rebound can occur rapidly following antigenic activation or treatment interruption [40]. Even intensification with ART has not been successful in eradicating the viral reservoir and preventing antiretroviral-free viral rebound [41]. Thus, eradicating the latent HIV reservoir has become a key scientific challenge in the development of vaccines and curative treatments for HIV [42,43,44]. Current strategies aimed at eliminating the latent HIV reservoir include several innovative approaches, which are as follows: (1) Early initiation of ART, in which HIV reservoirs can form and expand rapidly within days of infection. By promptly identifying newly infected individuals and initiating ART as early as possible, the initial establishment and expansion of viral reservoirs can be minimized [45]. (2) The “Shock and Kill” strategy, which is an approach that aims to reactivate latent HIV-infected cells, thereby allowing the expression of viral proteins, which can then be targeted and eliminated by the immune system [46]. Some clinical trials, such as the RIVER study, have explored the feasibility of this strategy, showing promising results in clearing latent reservoirs [47]. A recent study demonstrated that its HSV-vectored therapeutic vaccine was able to reduce the size of latent reservoirs [48]. (3) The “Block and Lock” strategy, which is a strategy that focuses on preventing the reactivation of latent HIV by locking it in an inactive state through the use of specific compounds that inhibit the transcriptional machinery necessary for HIV activation [49]. In summary, while significant progress has been made in controlling active HIV infection with ART, the challenge of eliminating latent reservoirs remains a critical barrier to a complete cure. Innovative strategies targeting these reservoirs are essential to achieving a functional or sterilizing cure for HIV. In addition, while viral persistence mechanisms primarily inform HIV cure research, understanding latency reservoirs is critical for optimizing PrEP durability in seronegative high-risk populations [50].

### 2.4. Lack of Animal Infection Models

Animal models can replicate and mimic the process of viral infection in vivo, providing preliminary evidence for the safety and efficacy of drugs and vaccines, and thus are indispensable for elucidating the mechanisms of pathogenesis, and for the development of therapeutics and vaccines [51,52]. However, due to HIV’s strict host specificity, it can only cause obvious AIDS clinical symptoms in humans. While some primates can be infected with HIV, they do not exhibit typical clinical symptoms. Consequently, there is a lack of appropriate animal models for HIV-related studies. Currently, simian immunodeficiency virus (SIV) or chimeric simian/human immunodeficiency virus (SHIV) infections in non-human primates, such as rhesus macaques, are used to model HIV infections in humans [52,53]. Despite the similarities between SIV and HIV, there are significant genetic and biological differences between the two viruses [54], as well as differences in the immune systems of non-human primates and humans. As a result, the immune responses observed in these models may not fully replicate those observed in HIV patients, complicating the extrapolation of findings from animal studies to human applications in pathogenesis, drug development, and vaccine design. In addition, there are significant differences in the course of disease, as well as pathological changes and clinical manifestations between SIV and HIV, making it insufficient to rely on a single animal model for studying HIV infections. Moreover, the cost and ethical considerations associated with the use of non-human primates are also important factors that must be taken into account. Scientists are actively seeking inexpensive and easily accessible small animal models for HIV research. In recent years, the BLT (bone marrow, liver, and thymus) humanized mouse model has been used to partially replicate HIV infection in vivo, facilitating the study of pathogenesis, drug screening, and gene therapy [55,56,57,58,59]. However, due to the development of graft-versus-host disease (GvHD) and its inability to fully replicate the human immune system [60], there remains a need to develop alternative animal models that are more suitable and effective in the development of biomedical interventions against HIV infections.

### 2.5. Lack of Correlates of Protection

Once infected with HIV, no one has been found to spontaneously eradicate HIV and completely recover to the pre-infection state, suggesting that the human immune system alone is unable to provide effective immune protection to cure HIV infections. The correlates of protection are potential parameters to assess the protective efficacy conferred by biomedical interventions against HIV, and thus they are critical to determining whether the intervention is effective against HIV infections. Unfortunately, we still lack a comprehensive understanding of which immune components and antigenic targets are pivotal in controlling HIV infection [61,62,63]. This knowledge gap has hindered progress in the development of immunoassays to accurately predict the protective effects of interventions in humans, as well as in innovating immunological tools capable of inducing durable protective immune responses. Due to the high variability of HIV and the complexity of the infection process, it remains challenging to evaluate the protective efficacy of interventions using a single biomarker, further complicating the assessment of preventive effectiveness. Therefore, researchers are exploring a wide range of potential immunoprotective indicators, including the breadth and magnitude of antibody responses (both neutralizing and non-neutralizing), T-cell activity, and cytokine profiles [64,65,66,67,68,69]. For example, the results from an animal experiment indicate that CXCR5+ CD8 T cells expand rapidly during chronic SIV infection and play a key role in controlling SIV progression, demonstrating their potential as indicators of immune protection [70]. Additionally, cyclic GMP-AMPase has been identified as an innate immune sensor for HIV, which is rapidly activated upon HIV infection and induces the production of type-I interferons and a variety of cytokines, thereby exerting a protective effect against HIV [71]. Despite ongoing efforts, no universal and reliable indicators have yet been established to fully measure protection efficacy against HIV. A more comprehensive understanding of the immune system’s response to HIV is urgently needed to identify and validate more effective and reliable immunoprotective indicators. Such insights could significantly improve our ability to assess the immunoprotective effects of various interventions, thereby guiding the development of more effective HIV prevention and treatment strategies.

### 2.6. Ethical and Biosafety Issues

Clinical trials are an essential step in the development of HIV biomedical interventions. However, because HIV is highly pathogenic and practically incurable, conducting clinical trials for biomedical interventions, particularly in live HIV infection research, carries unavoidable risks. For example, during the studies of an HIV vaccine, researchers found that although vaccination with an attenuated live SIV mutant vaccine can provide effective protection against wild SIV infection, this strategy is still prohibited from further research in the human clinical trials due to potential virus recovery mutations [72]. Studies involving human HIV cohorts have also shown that even highly attenuated strains of HIV can still lead to AIDS-associated symptoms [73]. These findings highlight safety concerns during the development of biomedical interventions against HIV. Significant gaps persist in assessing the safety and efficacy of HIV prevention products for special populations, particularly pregnant and lactating women. These gaps primarily stem from ethical concerns regarding potential risks to infants, which often preclude their inclusion in clinical trials [74]. While randomized, double-blind, placebo-controlled trials (RCTs) remain the gold standard for minimizing bias and ensuring robust results, their application in HIV prevention research requires careful ethical scrutiny. Traditional placebo-controlled designs—where control groups receive no active prophylaxis—are now considered ethically unacceptable in this field [75]. This shift reflects the widespread availability of highly effective antiretroviral-based prevention (e.g., PrEP), which establishes a clear ethical obligation to provide proven protection to all trial participants [76,77]. Consequently, contemporary HIV prevention trials must balance scientific rigor with ethical imperatives, often adopting alternative designs (e.g., active-controlled or superiority trials), which might hinder the development progress to some extent.

## 3. Advances of Biomedical Interventions for HIV Prevention and Control

HIV prevention and control is the most key component in achieving the goal of eliminating AIDS. Biomedical interventions are crucial for HIV prevention and control, and we herein summarize current biomedical interventions for HIV prevention and control, aiming to provide a reference for the development of innovative strategies against HIV infections (Table 1).

### 3.1. PrEP

The concept of PrEP was introduced in the mid-2000s as a strategy to reduce HIV transmission in individuals at high risk who are HIV-negative. This approach involves the regular use of antiretroviral drugs to lower the risk of HIV acquisition. The initial PrEP drugs, primarily oral nucleoside reverse transcriptase inhibitors (NRTIs) like Tenofovir/Emtricitabine (TDF/FTC) and Tenofovir Alafenamide/Emtricitabine (TAF/FTC), have demonstrated good safety and efficacy in numerous studies [79]. In 2012, the World Health Organization (WHO) endorsed daily oral TDF/FTC as a PrEP measure for HIV-negative individuals in serodiscordant relationships, marking a significant milestone in HIV prevention. In 2015, the WHO expanded its PrEP recommendations to include men who have sex with men (MSM) and people who inject drugs, further broadening the scope of HIV prevention efforts [93]. Despite these advances, adherence challenges with daily oral PrEP, as observed in trials such as VOICE and KPNC, underscored the need for more user-friendly options [94,95]. Additionally, the short duration of protection and the need for frequent medication can lead to side effects, such as gastrointestinal irritation and nephrotoxicity [96,97]. Consequently, researchers are intensifying their efforts to develop long-acting antiretroviral drugs and alternative routes of drug delivery, beyond daily oral intake, to sustainably inhibit viral replication, prevent the emergence of drug-resistant strains of the virus, and address issues of patient privacy and social stigma. For example, cabotegravir, an integrase strand transfer inhibitor given as an injection every two months, showed higher efficacy than daily TDF/FTC in trials (HPTN 083 and HPTN 084) involving MSM and transgender women [84]. This led the WHO, in 2022, to recommend cabotegravir as a long-acting PrEP option for individuals at a high risk of HIV infection. Nevertheless, two critical considerations persist with long-acting cabotegravir (CAB-LA) implementation, as follows: (1) the emergence of LA-CAB resistance mutations in breakthrough infections [98], and (2) the phenomenon of prolonged viral suppression with subsequent immunological discordance, which is termed long-acting early viral inhibition (LEVI) syndrome, following undiagnosed acute HIV infection at drug initiation [99]. Islatravir (ISL), a new class of nucleoside reverse transcriptase translocation inhibitor, is undergoing evaluation in the Impower 022 and Impower 024 trials to assess its effectiveness across different populations. Of note, lenacapavir, a long-acting capsid inhibitor, demonstrated a strong preventive effect against HIV transmission. A single subcutaneous injection of lenacapavir provided drug exposure for up to six months, leading to its approval for HIV prevention in the European Union in 2022 [100]. Importantly, impressive findings from the Phase III clinical trial (PURPOSE 1) recently revealed that administering lenacapavir subcutaneously every six months resulted in almost 100% efficacy in preventing HIV infections [85], effectively improving the PrEP dosing regimen and the single route of delivery, which is of great significance in meeting the goals of ending AIDS by 2030 [101]. However, it should be noted that the global distribution of newer antiretroviral agents like lenacapavir may face significant limitations in the near future. These constraints primarily stem from manufacturing capacity limitations, which contribute to both supply shortages and prohibitively high costs, thereby restricting accessibility in many regions. Here, we summarize those key Phase III PrEP clinical trials (Table 2).

The tissue distribution profiles of PrEP agents exhibit significant site-specific variations across different mucosal surfaces. Pharmacokinetic studies demonstrate that tenofovir and emtricitabine achieve substantially higher drug concentrations in colorectal tissues compared to vaginal mucosa [105]. This differential distribution results in suboptimal drug exposure at certain anatomical sites, particularly in vaginal tissue where the drug concentration may fail to reach the threshold required for effective prophylaxis. Consequently, the preventive efficacy of PrEP shows marked anatomical variation depending on the exposure site. Alternatively, the dapivirine ring represents a new PrEP-based approach that offers a long-term HIV prevention option for women who are unwilling or unable to take daily oral medication. Unlike oral PrEP, it does not require daily or pericoital use. The dapivirine ring is a silicone ring inserted into the vagina, which provides a sustained, localized release of dapivirine (a non-nucleoside reverse transcriptase inhibitor) over a 28-day period, offering women a controlled, long-lasting, and discreet form of HIV prevention. Several studies have demonstrated that the dapivirine ring has a favorable level of safety and is effective in decreasing the risk of HIV infection, and can have improved efficacy with improved adherence [82,106]. In 2021, WHO endorsed the dapivirine ring as an additional HIV prevention method for women at high risk [107]. However, the dapivirine ring still has several limitations, including the following: (a) it is only suitable for vaginal intercourse and cannot protect against other routes of HIV transmission; (b) its overall protection effect ranges from 27% to 63%, which may not provide sufficient protection; and (c) localized vaginal irritation may reduce adherence to its use. Additionally, research on the rectal dapivirine ring may further expand the scope and application of PrEP.

Long-lasting biomedical delivery devices, whether administered locally or systemically, are also being explored for PrEP, as they can improve the invisibility, effectiveness, and compliance of biomedical interventions [108]. Long-acting antiretroviral implants represent a promising and highly effective HIV intervention technology, providing sustained release of antiretroviral drugs for months or even years after being implanted under the skin. These implants offer a convenient, long-term solution for preventing HIV infection by suppressing viral replication. Several research teams are currently engaged in the development of long-acting antiretroviral implants and investigating their feasibility and safety [108,109,110,111,112]. Long-acting antiretroviral implants achieve sustained drug release through various modalities, such as the following: (1) refillable nanochannel implants that maintain therapeutic tenofovir alafenamide (TAF) concentrations for 83 days; [109] (2) biodegradable or non-biodegradable polymer matrices that provide islatravir (ISL)-based protection for six months [113]; and (3) poly lactic-co-glycolic acid (PLGA) bioerodible matrices that sustain effective dolutegravir (DTG) levels beyond five months [114]. However, challenges remain in stabilizing the drug release rate and mitigating potential adverse effects associated with the implants, such as inflammation [115,116]. Long-acting antiretroviral implants with controlled adverse effects and stably releasing drug concentrations up to the potent dose could contribute greatly to HIV prevention if they are successfully put into clinical use.

### 3.2. PEP

After HIV exposure via vaginal mucosa, the detection of HIV in regional lymph nodes can be achieved in about 2 days, and it takes 5 days for HIV to be detected in the blood, suggesting that there is a window of opportunity after HIV exposure [117]. PEP is the intervention aimed at blocking HIV infection during this window phase by administering antiretroviral drugs for 28 consecutive days, ideally starting within 24 h, and no later than 72 h after exposure. PEP provides an emergency, remedial HIV prevention option for individuals who have been exposed to HIV through an occupational or nonoccupational event. The evaluation of PEP efficacy presents unique methodological challenges due to its inherently episodic nature as a single-course intervention. The absence of RCTs assessing PEP effectiveness for HIV prevention reflects these difficulties, as the transient and unpredictable nature of exposure events complicates the design of robust clinical studies. Nevertheless, extensive preclinical evidence from animal models has consistently demonstrated the efficacy of PEP in reducing the HIV transmission risk [118,119]. These experimental findings are further corroborated by population-level studies documenting PEP’s effectiveness in real-world settings [87,120,121]. The convergence of these complementary lines of evidence has led to the widespread adoption of PEP in clinical practice, with regulatory approval granted in most national HIV prevention guidelines globally. Coformulation of zidovudine (ZDV) and lamivudine (3TC) was first recommended for occupational PEP [122], and later for nonoccupational PEP [123]. Contemporary guidelines generally recommend replacing zidovudine with tenofovir and using DTG as the preferred third drug for PEP in NRTIs-based dual or triple combination regimens [124,125,126]. As a remedial prophylactic measure, PEP does not provide 100% protection, and thus should not be relied upon as the sole or primary HIV prevention strategy. Reported failed reasons after PEP include delayed initiation, poor or non-adherence, and continued high-risk sexual exposures after PEP administration. It is worth noting that studies have shown that the convenience of PEP may lead at-risk populations to over-rely on it, substituting it for other preventive measures, which increases the risk of HIV transmission [127]. Additionally, the timing of PEP initiation and adherence are critical factors influencing its effectiveness [128]. The main risks associated with PEP include adverse reactions to antiretroviral (ARV) drugs and the potential emergence of drug-resistant strains of HIV. Consequently, several trials have investigated the adverse effects and tolerability of PEP (Table 3). The WHO guidelines advise that recurrent PEP users or individuals with ongoing HIV exposure risks should transition to PrEP for more effective, continuous prevention. Despite the risks of failed protection and the occurrence of adverse events, the use of PEP continues to grow, as it provides a critical emergency intervention following potential HIV exposure.

### 3.3. Treatment as Prevention (TasP) and U=U Principle

The WHO’s 2023 HIV Control Guidelines points out “Three-tiered framework of HIV control encompassing (1) population-level incidence reduction, (2) suppression of disease progression in seropositive individuals, and (3) functional cure/eradication research”. Through promoting current ART treatment and future functional cure/eradication treatment, AIDS patients (the source of HIV transmission) will be effectively controlled and reduced, so as to reduce and prevent the population-level incidence of HIV infections. TasP is a biomedical prevention strategy to decrease the risk of HIV transmission by controlling the viral load in PLWH through ART. U=U, a core scientific concept in the TasP strategy, explicitly states that as long as an HIV-infected individual has an undetectable viral load, the risk of sexual transmission of HIV is considered to be zero [143]. Numerous studies have shown that the risk of HIV transmission is directly correlated with the viral load in PLWH, and the possibility of HIV transmission can be substantially reduced when the viral load is lowered to very low levels through ART [144,145]. When the viral load reaches undetectable levels, PLWHs no longer exhibit AIDS-related symptoms and are considered non-infectious [143,146]. In fact, there is almost zero risk of sexual transmission of HIV when there are viral loads less than 1000 copies/mL [88]. The current findings indicate that U=U applies at an HIV RNA threshold of 200 copies/mL [147]. Based on the TasP strategy, the 2015 universal ART guidelines recommended that ART be initiated immediately after HIV diagnosis to facilitate viral suppression [148]. The TasP strategy, along with the U=U concept, was formally endorsed by the WHO in 2018, which emphasized that maintaining an undetectable viral load through adherence to ART is key to achieving both TasP and U=U.

In addition to ART for achieving U=U, recent research has explored innovative biological interventions to combat HIV. Among these, therapeutic interfering particles (TIPs)—defective viral particles that replicate conditionally in the presence of wild-type HIV—have emerged as a promising strategy [149]. TIPs act as “parasites”, depleting the resources necessary for wild-type HIV to replicate and proliferate. This competitive interference can continuously reduce the viral load of HIV and exert antiviral effects. Recently, a study has shown that single-dose HIV treatment with TIPs has the potential to limit the viral load and reduce HIV transmission by lowering the viral load to below the transmission threshold defined by the WHO [150], and thus meeting the U=U for HIV prevention. The development and widespread acceptance of these concepts have not only advanced our understanding of HIV prevention, but have also contributed significantly to reducing the stigma faced by PLWH, thereby slowing the global HIV epidemic.

### 3.4. Testing as Prevention

HIV testing represents a cornerstone of HIV prevention and control strategies, frequently discussed in conjunction with TasP. Despite progress, approximately 14% of people living with HIV remain unaware of their infection status, with many recognizing their infection only at advanced stages. Research indicates that delayed ART initiation due to late diagnosis significantly contributes to increased HIV transmission risk. The use of HIV rapid test kits enables early detection, helping to identify undiagnosed infections. By facilitating prompt ART initiation, these tests play a crucial role in achieving viral suppression, thereby reducing transmission rates. Modeling studies demonstrate that when HIV self-testing kits are introduced as a supplementary testing method among MSM, they could reduce HIV prevalence by 10%, while substantially improving awareness of serostatus within this population [151]. Furthermore, organized large-scale HIV testing initiatives (e.g., crowdsourced testing) in MSM communities have proven to be effective in lowering HIV incidence [152]. These findings underscore the indispensable role of HIV testing in prevention efforts, even though the concept of “testing as prevention” has yet to gain widespread recognition.

### 3.5. Prevention of Vertical Transmission

Prevention of vertical transmission is a strategy designed to prevent HIV transmission from mother to child, which is the primary route of HIV infection in children. The approach involves a range of intervention measures aimed at interrupting this transmission. Statistics indicate that, in the absence of preventive interventions, the cumulative rate of vertical transmission of HIV during pregnancy, delivery, and breastfeeding can be as high as 45% [153]. This probability can be reduced to 2% after intervention with ART, and can be further reduced (<1%) with a combination of intervention measures [154]. Research has demonstrated that the mother’s HIV viral load is a critical determinant of vertical transmission risk. When the mother’s viral load is undetectable, the likelihood of transmission to the child is virtually eliminated, which is consistent with the concept of U=U. In addition, continued ART during breastfeeding can further reduce the risk of HIV transmission through breast milk [154]. The WHO recommends that HIV-positive mothers in resource-limited settings breastfeed while maintaining effective ART, as the nutritional and immunological benefits for the infant substantially outweigh the low residual transmission risk when the maternal viral load is suppressed. However, it must be emphasized that ART-suppressed breastfeeding still carries a non-zero transmission risk. In settings where safe alternatives are available (including affordable, sustainable formula feeding), replacement feeding remains the preferred option to completely eliminate postnatal HIV transmission. In cases where the mother’s viral load is not adequately controlled at the time of delivery, the cesarean section can still significantly reduce the risk of transmission [155]. Current clinical guidelines recommend intravenous zidovudine administration during intrapartum care for pregnant women with HIV viral loads exceeding 1000 copies/mL at delivery. Furthermore, the duration and intensity of infant antiretroviral prophylaxis should be stratified according to transmission risk, which is primarily determined by the degree of maternal virological control throughout pregnancy and viral load status at delivery. Through the promotion and popularization of interventions to prevent vertical transmission in recent years, the global rate of mother-to-child transmission of HIV has been successfully controlled [3].

### 3.6. Voluntary Medical Male Circumcision (VMMC)

Several randomized controlled trials examining the effect of circumcision on HIV risk in heterosexual men have demonstrated that circumcised men experience significantly lower rates of HIV infection (approximately 60%) compared to their uncircumcised counterparts [90,91,92]. However, circumcision does not reduce the level of HIV in the genital tract or the likelihood that people living with HIV transmit the virus to their sexual partner. In fact, possible mechanisms for the role of VMMC in reducing HIV infection in men who engage in penetrative sex include the following: (1) Reduction of HIV-susceptible tissue. Large numbers of HIV target cells (such as CD4+ T cells and Langerhans cells) are concentrated on the inner mucosal surface of the foreskin [156,157]. Voluntary medical male circumcision (VMMC) removes this tissue, thereby reducing the surface area vulnerable to HIV infection. (2) Alteration of genital microbial composition and inflammation. VMMC may protect against HIV by modifying the genital microbiome and reducing inflammation. The procedure exposes the glans to a drier environment, decreasing the abundance of anaerobic bacteria and associated inflammation [158,159]. This reduces the likelihood of HIV encountering target immune cells and establishing infection. (3) Reduction in co-infections that facilitate HIV transmission. VMMC can lower the incidence of sexually transmitted infections (STIs) linked to HIV acquisition, such as human papillomavirus (HPV) and herpes simplex virus type 2 (HSV-2) [160]. By mitigating these biological co-factors, circumcision indirectly reduces HIV susceptibility. Long-term follow-up data further support the efficacy of circumcision in consistently reducing the risk of HIV infection [161]. Therefore, UNAIDS recommends VMMC as an HIV prevention measure in areas with a high HIV prevalence rate [162]. VMMC is not currently recommended to prevent HIV transmission in MSM and transgender women. Nonetheless, recent studies showed that VMMC also play a role in reducing the risk of HIV infection among MSM [163,164], possibly because VMMC may be of some effect in exclusively insertive sexual intercourse. Consequently, VMMC has proven to be an effective strategy for HIV prevention in men. Furthermore, while VMMC offers HIV prophylaxis to a certain extent, it does not eliminate the risk of infection, and should therefore be used in conjunction with other preventive measures to maximize its effectiveness.

### 3.7. Gene Editing Technology

Achieving an HIV cure through the body’s own immune system remains extremely difficult; however, there have been a small number of cases worldwide where HIV patients have been successfully cured. In 2009, the first reported case of an HIV cure through stem cell transplantation occurred in the so-called “Berlin Patient,” a person with both acute myeloid leukemia and HIV-1 infection. This patient was cured following a stem cell transplant from a donor with a 32-nucleotide deletion in the CCR5 gene (CCR5 delta 32), which confers resistance to HIV infection [165]. Since then, additional cases have been reported, including the “London Patient”, “Dusseldorf Patient”, “New York Patient”, “City of Hope Patient”, “Geneva Patient”, and another “Berlin Patient”, all of whom were cured of HIV after receiving stem cell transplants [166,167,168,169]. Furthermore, investigations have shown that some Caucasian individuals with the CCR5 delta32 mutation gene remain uninfected and inherently immune to HIV despite extensive exposure to HIV [170]. These findings highlight the potential for genetic approaches in both HIV prevention and treatment. Unfortunately, due to the invasive nature of stem cell transplantation and the stringent conditions under which it is performed, this approach is not universally applicable, and is primarily reserved for patients with severe co-existing conditions, such as malignant tumor. Therefore, researchers are actively exploring gene editing technologies as a promising and accessible alternative for HIV treatment and prevention.

Gene editing technologies, such as CRISPR-Cas9 and Zinc Finger Nucleases, primarily function by modifying either the host cell or viral genome. These technologies have emerged as a promising avenue for exploring novel biomedical interventions for HIV prevention. For HIV-infected individuals, natural immunity to HIV can potentially be achieved by knocking out or modifying HIV co-receptors, such as CCR5 or CXCR4, in the host cell [171,172]. Results from a clinical trial have demonstrated that transplantation of hematopoietic stem and progenitor cells with CCR5 knockdown into patients with HIV-1 infection and acute lymphoblastic leukemia by CRISPR editing can achieve a long-term reconstitution of the patient’s hematopoietic system without the occurrence of gene editing-related side effects, and can produce partial control of HIV infection during ART cessation. However, the percentage of CCR5-deficient lymphocytes following treatment remains relatively low (approximately 5%), which results in limited efficacy in controlling HIV infection, underscoring the need for further research to refine this approach [173]. For the viral genome, gene editing offers the potential to directly target and excise integrated HIV DNA, including from latent viral reservoirs, thus preventing HIV proliferation, or even achieving a cure [174,175]. HIV-specific engineered recombinases (e.g., Brec1) represent a promising gene editing approach for the targeted excision of integrated proviral DNA from the host genome. The molecule’s precise recognition of HIV long terminal repeats (LTRs) substantially minimizes off-target effects. In vitro studies demonstrated that Brec1 excised >90% of integrated proviral DNA, with no detectable off-target activity. Furthermore, in vivo experiments involving transplantation of Brec1-modified hematopoietic stem/progenitor cells (HSPCs) into humanized mice resulted in significant viral load reduction, indicating a potential for sustained immunological control of HIV infection. Notably, viral vectors have also been used as tools for gene editing techniques. This is done by using viral vectors, such as retroviral vectors (mainly lentiviral vectors), which remove pathogenicity but retain the ability to deliver genes, to insert anti-HIV gene fragments into the host cell so that they can be expressed efficiently and consistently in the host cell, thereby preventing HIV infection [176,177]. However, the application of gene editing technology to humans faces enormous ethical challenges. Furthermore, gene editing technology must overcome challenges such as off-targeting effects, HIV’s high variability, and the development of efficient delivery systems [178,179]. There are also concerns about the potential mutations by gene editing, thus leading to tumorigenesis [180,181]. Therefore, additional clinical trials are required to assess the safety and reliability of gene editing as a viable HIV intervention.

### 3.8. Passive Infusion of Broadly Neutralizing Antibody

With longitudinal co-evolution between the HIV variants and the B-cell lineages, the broadly neutralizing antibody (bnAbs) can be generated in about 1% of HIV-infected patients after multiple years [182,183,184]. Increasing HIV-specific bnAbs have being isolated, including VRC01, B12, 3BNC60, 3BNC117, 2F5, 4E10, 2G12,PG9, PG16, and CH01, etc. Broadly neutralizing antibodies (bNAbs) can potently inhibit both cell-free and cell-to-cell HIV transmission, effectively reducing viral infection of target cells across diverse HIV isolates. The breadth and potency of this neutralization vary significantly, depending on the bNAb’s epitope specificity and structural characteristics. Passive infusion of bnAbs has achieved not only a therapeutic effect in HIV patients, but also a potential for HIV prevention in humans. Of note, passive infusion of bnAbs also showed a role in the prevention and control of mother-to-child HIV transmission [185,186]. However, monotherapy with a single broadly neutralizing antibody (bNAb) demonstrates limited clinical efficacy, as exemplified by the extensively studied VRC01 antibody. Clinical trial data revealed that intravenous VRC01 administration every eight weeks provided protection solely against VRC01-sensitive HIV strains, while failing to reduce the overall HIV-1 acquisition rates compared to placebo [187], or to prevent viral rebound after ART interruption [188]. Consequently, there is an urgent need to develop next-generation bNAbs with both improved breadth against diverse HIV strains and an extended serum half-life. A promising strategy involves engineering multispecific bNAbs, which incorporate multiple antigen-binding domains into a single antibody scaffold. Preclinical studies demonstrate that such bispecific or trispecific bNAb constructs exhibit significantly enhanced antiviral breadth and neutralization potency in animal models compared to single bNAbs [189,190]. Furthermore, administration of combined bnAbs can further improve the broad-spectrum, durability, and efficacy against different HIV mutants [191,192]. In addition, the passive infusion of bnAbs has been demonstrated with acceptable safety and tolerability in humans [193], and it is expected to play a vaccine-like role in preventing HIV infection in the future.

Vectored immunoprophylaxis is an emerging strategy for HIV prevention, and different vectors (such as AAV) are used to express bnAbs or other anti-HIV factors that provide sustained protection against HIV in the host [194,195,196], making them a promising biomedical intervention for HIV prevention.

## 4. Conclusions and Prospective

We are currently in the most critical decade for achieving the 2030 goal of ending AIDS epidemic. A prophylactic vaccine is thought of as the most cost-effective biomedical intervention to eventually terminate the HIV epidemic among the general population, but it seems to not be clinically available soon due to technical issues. Though there is still no available HIV vaccine approved for clinical use, increasing biomedical interventions beyond vaccination, including through the use of PrEP, PEP, TasP/U=U, PMTCT, Dapivirine, and VMMC, is emerging for the prevention of HIV infections. Moreover, progress has been made in broadly neutralizing monoclonal antibodies, long-acting antiretroviral implants, immunoprophylaxis, interfering viral particles, and gene editing. These interventions are promising to intercept the risk of HIV acquisition and transmission, but many challenges are encountered during the promotion of these strategies in the real-world populations, such as prevention efficiency, which still needs improvement, adherence issues, and stigmatization due to long-term medication, risk compensation behaviors after intervention, etc. To end AIDS, there should be more out-of-the-box thinking. Overall, before achieving an effective HIV vaccine, ongoing research for developing alternative biomedical interventions offers a new hope for ending AIDS in the near future.

## Figures and Tables

**Figure 1 viruses-17-00756-f001:**
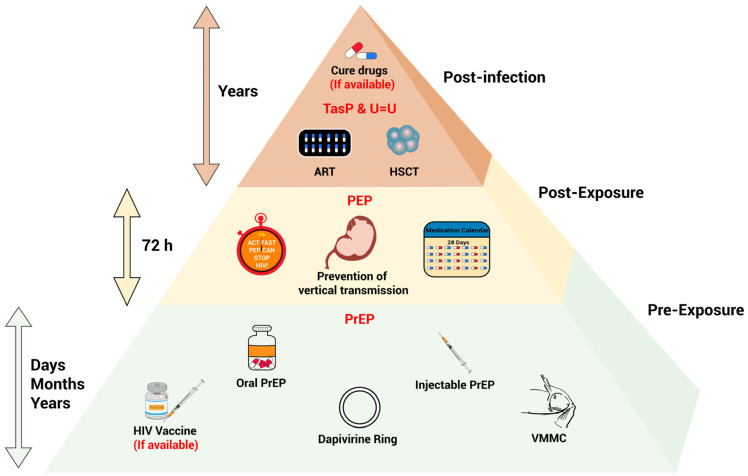
Biomedical intervention strategies for HIV prevention and control. All biomedical intervention strategies are categorized based on the different stages of HIV exposure. The different colors represent distinct stages of HIV exposure, ordered from bottom to top: pre-exposure, post-exposure, and post-infection. The arrows indicate the duration of each stage. In general, with more earlier stages of virus exposure, more intervention measures are available and effective against HIV infections. TasP and U=U: treatment as prevention and undetectable = untransmittable; ART: antiretroviral therapy; HSCT: haematopoietic stem cell transplantation; PEP: post-exposure prophylaxis; PrEP: pre-exposure prophylaxis; VMMC: voluntary medical male circumcision.

**Figure 2 viruses-17-00756-f002:**
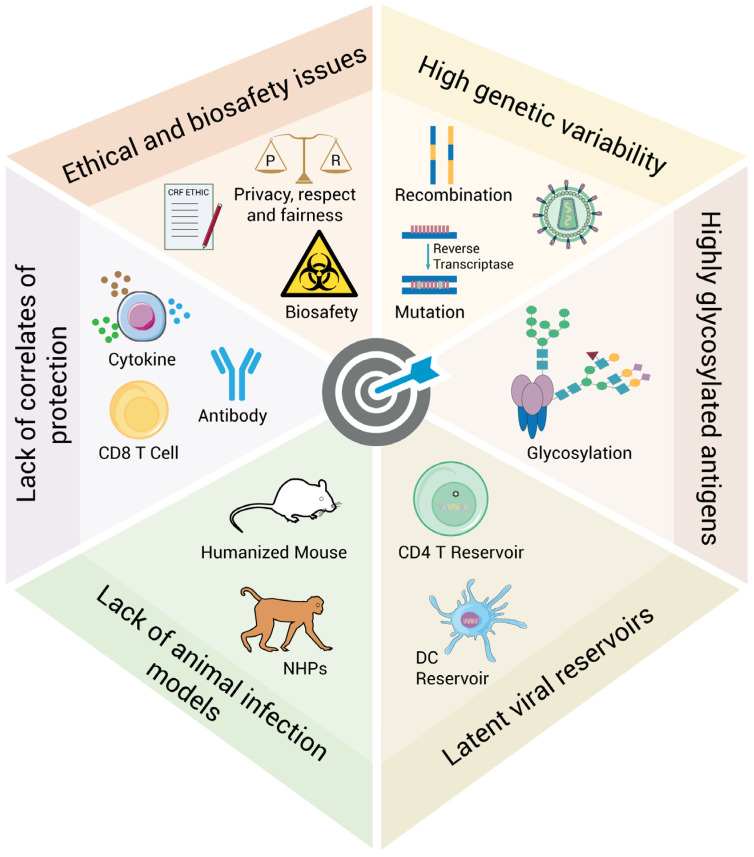
Challenges for the development of biomedical interventions against HIV infections. On the right half are the challenges posed by the characteristics of HIV itself, including high genetic variability, highly glycosylated antigens, and latent viral reservoirs. On the left half are the challenges created by the limitations of technologies, including the lack of animal infection models, and the correlates of protection, ethical, and biosafety issues. NHPs: non-human primates, DC: dendritic cell.

**Table 1 viruses-17-00756-t001:** Summary of current biomedical interventions for HIV prevention and control.

Category			Applicable Population	Prophylactic Efficacy	Advantage	Disadvantage
Currently available biomedical interventions	PrEP	Oral PrEP	Uninfected people who are persistently at high risk of HIV infection	Sexual transmission: 90–99% [78,79] Drug injection transmission: 74% [80]	1. Broad range of application 2. High efficacy for HIV prevention	1. Highly affected by adherence, stigmatization issues 2. Long-term medication, drug side effects
		Dapivirine ring	Uninfected women at high risk of HIV infection	Vaginal sexual transmission: 56–63% [81,82]	1. Female-led interventions 2. Long duration of protection 3. No need for oral medication, those females unable or unwilling to use oral medication can use this mode of prevention 4. Localized medication, reducing systemic side effects of medications 5. Strong concealment	1. Limited efficacy for HIV prevention 2. Limited to vaginal route of HIV transmission 3. Might cause local discomfort and inflammation
		Injectable PrEP	Uninfected people who are persistently at high risk of HIV infection	CAB-LA: 69–88% [83,84] Lenacapavir: 96–100% [85,86]	1. Long duration of protection 2. High efficacy for HIV prevention	1. Cost-related treatment barriers 2. Limited treatment availability 3. Drug resistance 4. LEVI syndrome
	PEP		Uninfected people who have been exposed to HIV or accidentally exposed	Occupational and nonoccupational exposure: 80–90% [87] (up to 99% for use within 2 h)	1. Fast-acting, can be used for emergency prophylaxis 2. Short-term use and convenient	1. Highly affected by initiation time and adherence 2. Drug side effects 3. Not applicable to people with persistent high risk
	TasP and U=U		All PLWH	Nearly 100% [88] (near elimination of risk of HIV transmission)	1. High prevention rate 2. Applicable to all people living with HIV 3. Reduce HIV stigma	1. High compliance requirements 2. Long-term medication, drug side effects
	Prevention of vertical transmission		HIV-positive women planning to become pregnant, pregnant, and breastfeeding	99% [89] (risk of mother-to-child transmission reduced to <1%)	1. High prevention rate 2. Improve the health of pregnant women and prevent newborns from being infected with HIV	1. Highly affected by adherence and stigmatization issues 2. Poor accessibility in underdeveloped regions where the need is greatest
	VMMC ^#^		Adolescent and adult males	Reduce the risk of heterosexual transmission of HIV infection by 60% [90,91,92]	1. No need to use drugs 2. Long-term protection without dependence on adherence 3. Reduce the risk of other sexually transmitted infections	1. Limited preventive effect, need to be combined with other measures 2. Ineffective in MSM and female partners 3. Surgical risk
Potentially available biomedical interventions in the future	Gene editing technology		–	–	1. Long-term protection without dependence on adherence 2. Removal of latent reservoirs for functional cure	1. Off-target risk 2. CCR5 receptor deficiency leads to susceptibility to other viral infections
	Passive infusion of broadly neutralizing antibody		–	–	1. Highly effective and broad-spectrum 2. Long-term protection without dependence on adherence 3. Both preventive and therapeutic	1. High production and preservation costs 2. Unable to remove latent viral reservoirs 3. Repeated infusions can trigger anti-drug antibody (ADA) responses

PrEP: pre-exposure prophylaxis; PEP: post-exposure prophylaxis; TasP and U=U: treatment as prevention and undetectable = untransmittable; PLWH: people living with HIV; VMMC: voluntary medical male circumcision; MSM: men who have sex with men; CAB-LA: long-acting cabotegravir; LEVI: long-acting early viral inhibition. ^#^ The protective effect of voluntary medical male circumcision (VMMC) among men who have sex with men (MSM) remains uncertain. Biologically, VMMC would primarily benefit those engaging in exclusively insertive intercourse, given the reduced risk of HIV acquisition through the penile mucosa. While partial evidence suggests that VMMC may confer some protection for MSM—particularly those practicing insertive sex—current data are insufficient to support broad recommendations. Notably, the WHO does not presently endorse VMMC as an HIV prevention strategy for MSM.

**Table 2 viruses-17-00756-t002:** Summary of key Phase III PrEP randomized clinical trials.

Trial	Timeline	Location	Population	Sample Size	Regimen	Outcome
iPrEx [78]	2007–2010	United States, Brazil, Ecuador, Peru, South Africa, Thailand	MSM and transgender women	2499	Daily oral TDF/FTC	36 infections in the TDF/FTC group, and 64 infections in the placebo group. Overall 44% reduction; up to 92% reduction with high adherence
TDF2 [102]	2007–2010	Botswana	Heterosexual men and women	1219	Daily oral TDF/FTC	9 infections in the TDF-FTC group and 24 infections in the placebo group. 62% reduction
Bangkok Tenofovir [80]	2005–2010	Bangkok, Thailand	People who use injection drugs	2413	Daily oral TDF	17 infections in the TDF group (0.35 per 100 person-years) and 33 infections in the placebo group (0.68 per 100 person-years). 44% reduction
Partners PrEP [79]	2008–2011	Kenya, Uganda	Heterosexual serodiscordant couples	4758	Daily oral TDF/FTC or TDF alone	17 infections in the TDF group (0.65 per 100 person-years), 13 infections in the TDF/FTC group (0.50 per 100 person-years), and 52 in the placebo group (incidence, 1.99 per 100 person-years). 75% reduction with TDF/FTC; 67% reduction with TDF alone
PROUD [103]	2012–2014	United Kingdom	MSM	544	Daily oral TDF/FTC	3 infections in the TDF/FTC group (1.2 per 100 person-years) and 20 infections in the placebo group (9.0 per 100 person-years). 86% reduction
IPERGAY [93]	2012–2014	France, Canada	MSM	400	On-demand TDF/FTC	2 infections in the TDF-FTC group (0.91 per 100 person-years) and 14 infections in the placebo group (6.60 per 100 person-years). 86% reduction
DISCOVER [104]	2016–2019	United States, Canada, Europe	MSM and transgender women	5387	Daily oral TAF/FTC vs. TDF/FTC	7 infections in the TAF/FTC group (0.16 infections per 100 person-years) and 15 infections in the TDF/FTC group (0.34 infections per 100 person-years). TAF/FTC was non-inferior to TDF/FTC
HPTN 083 [84]	2016–2020	United States, Latin America, Asia, Africa	MSM and transgender women	4570	Long-acting intramuscular cabotegravir (CAB-LA) every month vs. daily oral TDF/FTC	13 infections in the CAB-LA group (0.41 per 100 person-years) and 39 in the TDF-FTC group (1.22 per 100 person-years). 66% more effective than daily TDF/FTC
HPTN 084 [83]	2017–2021	Africa	Cisgender women	3224	Intramuscular CAB-LA every month vs. daily oral TDF/FTC	4 infections in the CAB-LA group (0.2 per 100 person-years) and 36 infections in the TDF-FTC group (1.85 per 100 person-years). 89% reduction with CAB-LA compared to TDF/FTC
PURPOSE 1 [85]	2021–2024	South Africa, Uganda	Cisgender adolescent girls and young women	5338	subcutaneous lenacapavir every 6 months vs. daily oral F/TAF vs. daily oral F/TDF	0 infections in the lenacapavir group (0 per 100 person-years), 39 infections among in the F/TAF group (2.02 per 100 person-years), and 16 infections in the F/TDF group (1.69 per 100 person-years). Significantly lower than other two groups (100% efficacy in preventing HIV infections)
PURPOSE 2 [86]	2021–2024	United States, Argentina, Brazil, Mexico, Peru, Puerto Rico, South Africa, Thailand	Cisgender men, transgender women, transgender men, and gender-nonbinary persons	3265	subcutaneous lenacapavir every 6 months vs. daily oral F/TAF vs. daily oral F/TDF	2 infections in the lenacapavir group (0.10 per 100 person-years) and in 9 infections in the F/TDF group (0.93 per 100 person-years). Significantly lower than other two groups (reduced overall risk of infection by 96%)

TDF: tenofovir disoproxil; FTC: emtricitabine; TAF: tenofovir alafenamide; F/TAF: emtricitabine-tenofovir alafenamide; F/TDF: emtricitabine-tenofovir disoproxil.

**Table 3 viruses-17-00756-t003:** Summary of key PEP clinical trials.

Trial	Location	Exposure	Sample Size	Regimen	Protection Rates	Adverse Reaction Rate
Kahn et al. [129] (2001)	United States	Nonoccupational	401	ZDV/3TC	78%	Nausea (52%), fatigue (44%), headache (24%), diarrhea (15%), and anorexia (12%)
Winston et al. [130] (2005)	Australia	Nonoccupational	385	ZDV/3TC vs. ZDV/3TC/NFV vs. TDF/3TC/d4T	75% vs. 68% vs. 85%	Transaminase elevation (11% vs. 9% vs. 19%), diarrhea (6% vs. 51% vs. 25%), fatigue (39% vs. 32% vs. 30%), headache (17% vs. 12% vs. 1%), and nausea (81% vs. 42% vs. 23%)
Mayer et al. [131] (2008)	United States	Nonoccupational	371	TDF/FTC vs. TDF/3TC vs. ZDV/3TC	72% vs. 87% vs. 42%	Diarrhea (47% vs. 31% vs. 10%), fatigue (30% vs. 28% vs. 39%), nausea (22% vs. 19% vs. 56%), headache (22% vs. 19% vs. 25%), and dizziness (20% vs. 16% vs. 5%)
Tosini et al. [132] (2010)	France	Nonoccupational and occupational	249	TDF/FTC/LPV-r	67%	Diarrhea (80%), asthenia (66%), and abdominal pain (44%)
Diaz-Brito et al. [133] (2012)	Spain	Nonoccupational	200	LPV-r vs. ATV	64% vs. 64%	Gastrointestinal (70% vs. 41%), neuropsychiatric (11% vs. 16%), asthenia (17% vs. 23%)
McAllister et al. [134] (2014)	Australia	Nonoccupational	120	RAL/FTC/TDF vs. FTC/TDF	92% vs. 91%	Fatigue (37% vs. 26%), nausea (24% vs. 18%), abdominal cramps (21% vs. 12%), myalgias (9% vs. 0%)
Leal et al. [135] (2016)	Spain	Nonoccupational	243	TDF/FTC/LPV-r vs. RAL	66% vs. 80%	Gastrointestinal (57% vs. 58%), neuropsychiatric (14% vs. 23%), and asthenia (18% vs. 18%)
Leal et al. [136] (2016)	Spain	Nonoccupational	237	TDF/FTC/LPV-r vs. TDF/FTC/MVC	56% vs. 68%	Gastrointestinal (56% vs. 58%), neuropsychiatric (15% vs. 20%), and asthenia (19% vs. 18%)
Fatkenheuer et al. [137] (2016)	Germany	Nonoccupational and occupational	305	DRV-r vs. LPV-r	94% vs. 90%	Diarrhea (30% vs. 52%), nausea (16% vs. 28%), fatigue (13% vs. 18%), sleep disorder(0% vs. 4%)
Valin et al. [138] (2016)	France	Nonoccupational	234	FTC/TDF/ELV/COBI	92%	Fatigue (26%), nausea (25%), diarrhoea (17%), abdominal cramps (16%)
Milinkovic et al. [139] (2017)	United Kingdom	Nonoccupational	213	TDF/FTC/LPV-r vs. TDF/FTC/MVC	65% vs. 71%	Nausea or vomiting (39% vs. 30%, diarrhea (74% vs. 19%), fatigue (39% vs. 36%)
Chauveau et al. [140] (2019)	France	Nonoccupational and occupational	158	TDF/FTC/RPV	86%	Fatigue (35%), nausea (22%), diarrhea (20%), abdominal cramps (16%), headache (11%)
Nie et al. [141] (2021)	China	Nonoccupational and occupational	297	ABT/DTG vs. ABT/TDF/3TC vs. DTG/TDF/3TC	64% vs. 64% vs. 64%	Dizziness (7% vs. 7% vs. 7%), diarrhea (8% vs. 6% vs. 2%), asthenia (5% vs. 4% vs. 5%), and triglycerides increase (4% vs. 2% vs. 7%)
Liu et al. [142] (2022)	China	Nonoccupational	108	BIC/FTC/TAF	96%	Creatinine elevation (4%), headache (2%), diarrhea (2%), and nausea (1%)
Lacombe et al. (2024)	France	Nonoccupational	226	DOR	–	–

ZDV: zidovudine; 3TC: lamivudine; NFV: nelfinavir; TDF: tenofovir disoproxil; d4T: stavudine; FTC: emtricitabine; LPV-r: ritonavir-lopinavir; ATV: atazanavir; RAL: raltegravir; MVC: maraviroc; DRV-r: ritonavir-boosted darunavir; ELV: cobicistat-boosted elvitegravir; COBI: cobicistat; RPV: rilpivirine; ABT: lamivudine; DTG: dolutegravir; BIC: bictegravir; TAF: tenofovir alafenamide; DOR: doravirine.

## Data Availability

No new data were created or analyzed in this study.

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
