# Peer review of "Biomedical Interventions for HIV Prevention and Control: Beyond Vaccination"

_viruses, 2025, doi:10.3390/v17060756_

Round 1
Reviewer 1 Report
Comments and Suggestions for Authors
Liao et al. have provided a narrative review on strategies for HIV transmission prevention in spite of still lacking vaccination options. The work gives a nice overview on the subject. I have a few recommendations on how the work should be further improved before publication is considered.
- General comment: Considering the authors “thinking-out-of-the-box” approach, they should at least mention the “testing-as-prevention” option as an approach of HIV-transmission prevention. This approach effects both mutual testing prior to potentially risk sexual encounters as well as general low-threshold availability of HIV testing options. In Germany, for example, good experience has been made with commercially available HIV rapid tests in drug stores. Although broad studies on the “testing-as-prevention” concept are lacking, there are some promising modelling approaches suggesting potential beneficial effects of this preventive strategy. Therefore, “testing-as-prevention” should not be missing in a comprehensive review on HIV transmission prevention as provided by the authors.
- Major comment: Line 97: It is an oversimplification to state that HIV-2 was only endemic in local areas. In fact, HIV-2 transmission is much less efficient compared to HIV-1, however, low proportions of HIV-2 have to be globally expected due to migration and internation travel.
- Major comment: Table 1, last row: To avoid an oversimplification, the authors should clarify at least in a footnote that lacking effectiveness of VMMC in MSM primarily refers to receptive sexual intercourse, while at least some effect is likely in case of exclusively insertive sexual intercourse.
- Major comment: Lines 399-401: As they authors are certainly aware of, there is ongoing scientific debate on the residual risk of HIV transmission by breast feeding in spite of antiretroviral therapy, considering aspects like the transmission of cell-bound proviral DNA and the immature enteric barrier in newborns. The authors should at least mention this residual risk in order to avoid the misleading impression that breast feeding under antiretroviral therapy would be entirely safe. As the authors are certainly aware of, WHO recommendations for breast feeding in spite of HIV for certain regions of the world are based on balancing of residual HIV transmission risks and risks associated with undernutrition.
- Sub-heading gene editing technology: The authors focus on the CRISPR-CAS-approach approach and mention its limitations. For the sake of comprehensiveness, I suggest that they should at least also mention and shortly discuss the alternative approach using HIV-specific designer recombinases like Brec1. As the authors are certainly aware of, the recombinase approach has some advantages compared with the CRISPR-CAS approach, although its clinical effect and the safety of its application are still uncertain.
- Minor comment: Line 295: I assume, the authors mean “a strong preventive effect against HIV transmission” rather than a “strong preventive effect in people living with HIV”, don’t they? Please rephrase to avoid misunderstandings by the readers.
- Minor comment: The authors should make sure that non-scientific phrases are replaced throughout the manuscript. Terms like “HIV epidemic is ravaging” (first sentence of the introduction) are interpretations rather than descriptions and should be replaced by more balanced wording.
- Minor comment: Although I am not a native speaker myself, some phrases and Grammar variants read non-idiomatic. For example, confusion of adjectives and adverbs forms repeatedly occurs throughout the manuscript (e.g., “high genetic variability” instead of “highly genetic variability” in sub-heading 2.1.). Thorough proof-reading either by the authors or the journal is recommended in order not to weaken the manuscript’s impact by poor readability.
- Minor comment: Abbreviations should be spelled out at first use. This rule should also be applied for figures and the graphical abstract, because these elements should be understandable by themselves without referring to the manuscript text.
- Minor comment: The authors should adapt the citating style to the journal requirements.
- Minor comment: Figure 2, left triangle: The word “protection” should be written without separating the letter “p”.
12. Minor comment: Line 96: “HIV can be divided into HIV-1 and HIV-2.” Please rephrase this over-simplified statement using appropriate phylogenetic terms.
Comments on the Quality of English LanguagePlease see details in the comments for authors.
Reviewer 2 Report
Comments and Suggestions for Authors
The paper is a extensive review addressing the issue of biomedical interventions against HIV infection. However, it is a little bit confusing to me the term “prevention” in the title as the manuscript seems to go far beyond HIV prevention options as it explores the existing gaps in the HIV cure as well (as the paragraph 2.3 seems to be) along with gene editing (paragraph 3.6). May be “against HIV” is more suitable
- the first pages (2-8) are general overview on HIV immunology
- Paragraph 3 (pag 8, line 257): this paragraph refers correctly to “HIV prevention “
- In the table 1, the PReP section misses Long Acting options - namely cabotegravir LA (im) and lenacapavir LA (sc) – with relative advantages (adherence…) as the table quotes only oral PrEP (LA options are mentioned only in table 2).
- Table 2 (list of phase III clinical trials, add RANDOMIZED in the line) should also mention the route (im vs sc) and the timing of administration (1 vs 6 months) for LA Cabo and lenacapavir. The table 2 presents the final results in term of percentage of infection reduction, which is not informative: we need absolute number (there are 100 or 20 or 1 infection only?). Also follow up time is missing
- There is no mention of concerns for emerging resistance at failure with LA Cabo and LEVI syndrome associated to acute HIV infection during LA Cabo use.
- An additional table would be useful for investigational approach (such as ISLATRAVIR… or bNAbs) to present the future options)
- Pag 11, line 323: there is no mention of which medications have been exploring for long term and how long is the effect supposed to last
- Pag 14, paragraph 3.4: mention that clinical trials investigating U=U had a HIV RNA thresholds of 200 copies/mL
- Pag 14, paragraph 3.4: quote the Open window currently left for “breast feeding” in subjects who desire that option when HIV is fully suppressed in mothers
- Pag 15: spelling error
Reviewer 3 Report
Comments and Suggestions for Authors
Summary
This review aims to summarize the broad field of HIV prevention beyond HIV vaccines, defining challenges that make HIV a difficult pathogen to protect against and the solutions that have been developed.
General Comments
It is not clear what gap in knowledge this review is targeting compared to other contemporary reviews on HIV prevention technologies. The manuscript in its current form struggles with scope, straying into eradication of HIV following infection, and omitting obvious prevention strategies.
The challenges presented are highly mechanistic and would appear most relevant to HIV eradication after infection. They do not include demonstratedly important factors such as the stigmatization of people with HIV, denial of HIV, economic limitations, and the social barriers to adoption of proven effective interventions, nor structural barriers that focus the HIV epidemic in low resource areas and populations.
The most effective and powerful recent developments, long-acting injectable PrEP, are mentioned in the text, but they appear to be an afterthought, not incorporated into the graphics or tables. Longstanding approaches such as treatment of other STIs or prevention with barrier methods or other interventions such as doxyPEP is not discussed.
Many references are outdated or missing.
The language around HIV does not reflect current recommendations; the authors would do well to revise the wording based on modern language in the field. See, for instance, the UNAIDS terminology guidelines (https://www.unaids.org/en/resources/documents/2024/terminology_guidelines).
Specific Comments
Introduction
Citation 2 is not an appropriate summary of the state of HIV vaccine and cure science, but a limited third party report (by the authors) on a specific germline-targeting vaccine.
Lines 45-46: this sentence should be re-worded
The term “Non-AIDS events” is used somewhat confusingly – it is not typically used to signify risks associated with ART, but the text and references imply this unusual usage here.
The grouping of interventions in figure 1 is somewhat unclear. VMMC is a one-time intervention that acts over time but is listed as “daily prevention.” Similarly an HIV vaccine would not be likely to be necessary daily, but is included here. “Orally PrEP drugs” should perhaps be reworded to “Oral PrEP.” Injectable cabotegravir and lenacapavir should be included. A key component of PMTCT (preferred term is “prevention of vertical transmission”) is through treatment of pregnant individuals (i.e. U=U). Why “HIV Cure Drugs” are listed in the timeframe of “AIDS” is very unclear.
Figure 2 has formatting and English language issues.
Highly genetic diversity
This section focuses too much on the molecular mechanisms of HIV genetic diversity rather than the challenges it prevents to non-vaccine prevention. Aside from bnAbs, the majority of preventative interventions are neutral to HIV genotype.
The pseudodiploid description is unclear and the delivery of multiple RNA templates into an infected cell should be more explicitly described.
The cited reference estimates RT template switching 4-5 times per genome, not per gene as described in the current manuscript.
The description of HIV-1 and 2 and groups (separate zoonotic introductions) after the description of recombination is misleading. HIV-1 has evolved along 4 different lineages hypothesized to originate from separate introductions, not because of mutational constraint.
The contribution of APOBEC and other host mutagens should be included in discussions of the origin of HIV diversity.
Highly glycosylated antigens
Again, the relevance of this section to non-vaccine prevention is limited.
Glycan arrays are not a distinctive feature of HIV in that they do not distinguish it from other enveloped viruses (as mentioned in the following sentence in the text).
It is confusing to say that glycans “envelope” HIV (which has its own envelope proteins). Recommend rewording.
Latently viral reservoirs
Most importantly, why is a lengthy discussion of viral reservoirs important in a review on HIV prevention?
The nature of the reservoir (free cDNA versus integrated) is somewhat technical and appears to vary based on the timing and efficacy of ART. Recommend simplifying this section.
Citation 46 is not an appropriate reference to summarize host/ARV activity against latent virus.
Conventional ART does not clear virions, but prevents them from productively infecting cells.
To what is “Conventional vaccines” on line 156 referring Citations 49-51 are to basic investigations of SIV reservoir eradication (performed by the authors) and not appropriate references for the broader goals of HIV eradication efforts in the referring text.
While broadly accepted, citations are requested for the latency elimination strategies, particularly including the many studies evaluating the strategies mentioned.
Intensification with ART has NOT been successful in eradicating the viral reservoir, see: 10.1038/s41598-020-79002-w, https://doi.org/10.1172/jci171554
Lack of Animal Infection Models
This section repeatedly makes reference to the use of animal models for vaccines or therapeutics, but does not make a case on their necessity for prevention modalities.
The SIV/NHP references are 10+ years old.
The references to the use of BLT mice in HIV research are all > 5 years old. Citations to more recent studies are requested.
Lack of correlates of protection
“Correlates of protection” typically refers to a surrogate measure in a vaccine study. What is its relevance for the non-vaccine protective strategies this review focuses on?
HIV is not the only virus that persists in its host, and this claim is perplexing.
It is unclear in this section whether the protection being evaluated is against infection with HIV in the first place, immune dysregulation in the setting of HIV infection, or immune progression/AIDS.
The references for potential “immunoprotective indicators” are either very old (71) or very specific, not a broad overview of the field of HIV infection resistance, long-term non-progression, or control (see concern above that the focus of this section is not clear).
How is the expansion of CXCR5+ CD8 cells in SIV infection relevant to HIV protection?
Ethical and biosafety issues
The references in this section are decades old and do not reflect the current strategies for HIV vaccination or prevention research. There are entire journals devoted to the ethics of contemporary HIV research generally and prevention more specifically. For instance, in the current setting of highly effective antiretroviral PrEP, are additional placebo-controlled trials ethical?
Table 1
This table is markedly incomplete even within the topics discussed with in the manuscript. It lacks references to the wide variety of studies contributing primary data to these interventions. Injectable PrEP with cabotegravir/rilpivirine or lenacapavir has been entirely omitted. Barrier protection for sexual intercourse and needle exchanges for injection drug use are also well-studied public health interventions (albeit with their own limitations) that should be included. Serosorting is a behavioral practice which has also been studied.
PrEP
It is also important to note that the efficacy of PrEP varies based on the mucosal site of exposure, likely due to differing thresholds for protection at the vaginal, penile, and rectal mucosa.
The cost and availability of newer agents such as lenacapavir are significant barriers to their global adoption and should be mentioned.
PEP
On what measure is PEP judged “not yet widely used?”
Recommend referencing the WHO guidelines in addition to USA and UK.
The outdated discussion of overuse of PEP (from 1999) should be replaced by the contemporary practice of “rolling over” PEP to PrEP for individuals with ongoing exposure.
More modern references on PEP efficacy are also requested.
TasP and U=U
Reference 135 is not to treatment guidelines as the text suggests
PMTCT (recommend re-titling)
Citation is requested for the transmission rate of 45% without intervention.
A large numbers of studies have evaluated maternal viral load and the risk of transmission which are appealed to without reference as “research has demonstrated” (lines 396-398). Citations are requested.
This section lacks mention of several key interventions that are standard of care:
- IV zidovudine during delivery in cases of detectable maternal viral load
- Antiretroviral prophylaxis/presumptive treatment for infants based on risk of transmission
VMMC
The biologic protective mechanism of penile circumcision in reducing the risk of HIV acquisition is not known, and this should be stated alongside the hypothesis described.
It is also important to note that circumcision has not been universally shown to be protective against HIV, particularly in high-income regions.
bnAbs
Discussion of the AMP trials (HPTN 085, 081) is important when discussion antibodies for prevention – VRC01 infusion did not reduce overall HIV-1 acquisition over placebo.
The references in this section do not represent the latest research in this area.
Others
TIPs for treatment are arguably not a distinct mechanism of prevention from the TasP described previousy.
Conclusion and prospective
“[T]he spread of AIDS” is not a scientifically or medically accurate phrase.
Comments on the Quality of English LanguageSee above for recommendations on specific resources for improving the language discussing HIV. In general adverbs are frequently used in place of adjectives.
Round 2
Reviewer 3 Report
Comments and Suggestions for Authors
This version of the manuscript has amended misleading statements, largely modernized the language relating to HIV, included more modern approaches to HIV prevention, clarified some technical and grammatical details, and revised some of the inappropriate references.
The revised title and introduction remain somewhat confusing in why these aspects of HIV (prevention and control) are being discussed together, and the manuscript seems to suffer from confusion on this issue. Does “control” refer to the population level reduction in new infections, preventing disease progression within an individual, or complete eradication of HIV within a person? The selection of strategies that are discussed still does not reflect the diversity of strategies currently available to prevent HIV, let alone encompass all of the proven or developing strategies for within-host prevention of progression or eradication. The reasoning for combining these two strategies (historically investigated, funded, and implemented by different groups and using different approaches) is not provided, and I think the manuscript suffers for it. There is a strong emphasis on molecular mechanisms of viral persistence that has no relevance to HIV prevention and minimal impact on several aspects of HIV cure. Sections highlighted as minimally relevant to the review have been expanded without comment in the text or response to reviewers on why they need to be included.
Despite the clever graphical representation of the challenges vs interventions as logically related, the text presents them separately, with minimal connection between sections 1 and 2.
The majority of the references remain very old (>20 years), and this has not changed substantially in this revision despite the addition of some more modern literature. While historical detail is useful, a review summarizing publications from the early era of HIV is of limited utility in the current time, with modern treatment and prevention strategies.
Specific suggestions:
Graphical abstract has spelling and grammatical errors
Introduction:
If the entire population had access to appropriate ART, HIV infection would be prevented despite viral reservoirs by the principle of U=U, particularly with the addition of PrEP.
Individual antiretroviral agents are associated with cardiovascular disease or neurocognitive changes with prolonged use, but modern ART itself is not widely understood to be the source of increased cardiovascular disease or dementia in patients with treated HIV. Furthermore, whether these events would not qualify for the pharmaceutical definition of “serious adverse events.” Recommend more recent citations related to this, such as the REPRIEVE trial (10.1056/NEJMoa2304146).
Figure 1:
The inclusion of VMMC and HIV vaccine as “daily prevention” and distinct from “pre-exposure” remains unclear. In the case of vertical exposure, the virus and drug are (in the best circumstance) encountered at the same time, in utero with extended prophylaxis for the infant following exposure. It is typically likened more to PEP, but is obviously a more complicated scenario. Its inclusion in the category of “pre-exposure” does not seem appropriate. Safe sex promotion (e.g. codom use) is appropriately included here but not discussed in the text.
Latent viral reservoirs
Despite including citations that ART intensification with INSTI or entry inhibitors is ineffective, the strategy is still listed as under investigation.
Gene therapy (particularly related to co-receptors) is not relevant to viral reservoirs.
Lack of correlates of protection
The correlate being described remains unclear. There are direct assays for HIV infection (i.e. anti-HIV antibodies, RT-PCR for HIV genome in the plasma or from cells), so a surrogate marker is not necessary to evaluate prevention from infection. Progression of HIV disease is also routinely assessed via CD4 count, which has defined the stages of HIV infection for decades. This is perhaps a confusion arising from the conflation of preventing HIV infection and progression throughout the manuscript?
VMMC
The first two sentences of this paragraph state a hypothesized mechanism of the mechanism of HIV infection via the penile mucosa that has not been definitively established in the literature. Recommend re-wording and including more robust discussion. See, for instance:
10.1111/j.1600-0897.2010.00934.x.
10.1007/s11904-022-00634-w
The wording of “VMMC is not currently recommended to prevent HIV transmission in sex between MSM and transgender women” is somewhat unclear. It would be clearer to omit “sex between.”
The current discussion emphasizes the differences in recommendations based on sexual activity, but the recommendations are limited to regions with generalized epidemics (ie, high HIV prevalence) compared to North America and Europe.
Conclusion and prospective
“[T]he AIDS pandemic” would be better expressed as “HIV pandemic” or perhaps (based on historical usage, “To end AIDS.”
